# Youth Tobacco Control in the Digital Age: Impact of South Carolina’s Youth Tobacco Education and Vaping Cessation Social Media Programs

**DOI:** 10.3390/ijerph22020269

**Published:** 2025-02-12

**Authors:** Carolyn A. Stalgaitis, Susan Dang, Catherine Warner, Sharon Biggers, LaQuanna Jackson, Jeffrey W. Jordan

**Affiliations:** 1Rescue Agency, 2437 Morena Blvd, San Diego, CA 92110, USA; sdang@rescueagency.com (S.D.); jeff@rescueagency.com (J.W.J.); 2Division of Tobacco Prevention and Control, South Carolina Department of Public Health, 2100 Bull St, Columbia, SC 29201, USA; warnerca@dph.sc.gov (C.W.); biggersr@dhec.sc.gov (S.B.); jacksolj@dhec.sc.gov (L.J.)

**Keywords:** social media, adolescent, vaping, smokeless tobacco, tobacco use, tobacco control, tobacco cessation

## Abstract

To maintain relevance, youth tobacco control programs must leverage popular social media platforms and address evolving behaviors. Recognizing this, the South Carolina Department of Public Health and Rescue Agency implemented culturally tailored social media campaigns (*Down and Dirty*, *Fresh Empire*), a broad vaping social media campaign (*Behind the Haze*), and an Instagram-based vaping cessation program (*Quit the Hit*, *QTH*). This study examines program impact. The social media campaigns were evaluated via online cross-sectional surveys in 2019–2023. Analyses examined awareness and reception overall and among target audiences and compared knowledge, attitudes, and beliefs between campaign-aware and unaware participants. The impact of *QTH* was assessed via online baseline and follow-up surveys in 2021–2023. Analyses examined program feedback and changes in cessation confidence and tobacco use from baseline to follow-up. Over one-third of participants recalled the social media campaigns, and recall of featured facts was higher among the campaign-aware participants than the unaware participants. *QTH* participants’ confidence in quitting increased from baseline to follow-up, while any 30-day vaping and average number of days of vaping decreased. This innovative social media program reached high-risk youth with educational content and empowered teens to quit vaping, providing a model for comprehensive youth tobacco control programs.

## 1. Introduction

Following years of declining youth tobacco use in the U.S., the introduction of vapes led to a rapid increase in youth nicotine use [1,2,3,4,5]. While early advertising ostensibly targeted smokers wishing to quit cigarettes, the evolution of sleek, flavored pods and disposable vapes likely contributed to their adoption among youth [6,7]. Vapes have been the most commonly used tobacco product among youth since 2014, peaking at 27.5% of high school students in 2019 and declining to 10.0% in 2023 [8,9]. At the same time, youth continue to use other tobacco products, including cigarettes (1.9%), cigars, little cigars, and cigarillos (CLCCs; 1.8%), and smokeless tobacco (SLT; 1.5%) [8]. As such, a comprehensive youth tobacco control program must address a range of products to avert the health effects attributable to combustible and SLT use, including cancer, heart disease, and stroke, as well as the impacts of nicotine vaping on adolescent brain development and mood [10,11,12,13,14,15,16]. Given that most adults who use tobacco initiated in adolescence or young adulthood and early cessation reduces risk [17,18], efforts to prevent and reduce youth tobacco use are a worthwhile use of public health funding.

Many U.S. states supplement national youth tobacco control efforts with local programs. In 2018, the South Carolina Department of Public Health (SC DPH; previously South Carolina Department of Health and Environmental Control) partnered with Rescue Agency (Rescue), a behavior change marketing agency, to redevelop the state’s youth tobacco control program to address elevated youth tobacco use in the state [19]. The ensuing approach leveraged social media as its primary channel. Social media use in health education has grown immensely over the past two decades, allowing practitioners to leverage advantages, such as increasing equity by reaching audiences that are harder to reach with traditional mass media, using platform targeting techniques to deliver tailored messages to specific audiences, engaging the audience in two-way interactions rather than one-way mass media, fostering peer support, using multimedia such as videos to capture attention and overcome health literacy barriers, and decreasing costs while increasing a campaign’s reach compared to traditional media such as television [20,21,22,23]. However, evidence of the impact of social media campaigns on health behavior is mixed, and practitioners must grapple with potential downsides such as a lack of control over how a message is portrayed when it is reshared through the audience’s social network; copious misinformation that can drown out accurate health messages; privacy concerns surrounding sensitive health discussions occurring online; uneven access, which may contribute to inequities for certain populations such as the elderly; and potential exposure to risky behaviors or negative effects on mental health associated with social media use [20,21,22,23,24]. In the case of SC DPH’s program, a social media-centric approach was indicated, given that over half of teens, the program’s target audience, use social media daily [25]. Further, social media is a common source of teens’ exposure to pro-tobacco marketing and user-generated content, which may contribute to initiation [26,27,28,29,30,31,32], allowing the SC DPH’s program to interrupt pro-tobacco content teens encounter online. Additionally, the opportunity to foster peer support and engagement via social media was particularly promising for addressing teens’ cessation needs, as many young people looking to quit nicotine express a desire for social support [33].

### 1.1. SC DPH Youth Tobacco Control Program

SC DPH’s new programming was built upon the idea that the tobacco products used by youth and their reasons for use vary, so no single approach can effectively impact all teens. Instead, SC DPH and Rescue examined which teens in SC were at highest risk, their unique reasons for use, their messaging preferences, and their obstacles to behavior change. As a result, SC DPH diversified its program from a single general education campaign to include culturally tailored social media campaigns, a broad vape social media education campaign, and a vape cessation program delivered via Instagram.

This approach used Rescue’s *Decision Blocks*™ Strategic Framework to identify each audience’s specific blocks to behavior change and select the appropriate tactics to help them overcome those blocks [34,35]. Based on the Transtheoretical and Elaboration Likelihood Models (TTM, ELM) [36,37], the framework includes four common “blocks” to behavior change and corresponding tactics based on the audience’s stage of change. The first, Information Blocks, includes gaps in foundational knowledge about risks that are often observed in the TTM’s early stages of change [36]. Tactics for this block provide novel information or reframe existing knowledge and draw on the ELM to trigger peripheral processing to create an openness to the message [37]. Next are Impact Blocks, where the audience mentally distances themselves from known risks. Tactics for Impact Blocks leverage central processing to personalize risks and connect the message to the audience’s values and experiences, increasing readiness to change [36,37]. The third, Solution Blocks, occurs for audiences in the Action stage of the TTM when they are ready to change their behavior but struggle to picture themselves achieving that change [36]. Solution Blocks are countered by providing realistic, specific solutions to logistical barriers and presenting attainable changes and empowering tools. Finally, Reinforcement Blocks occur for audiences in the Maintenance stage of the TTM and are addressed by promoting sustainable support systems, suggesting maintenance strategies, and normalizing relapse and trying again [36]. When designing a *Decision Blocks* campaign, primary and secondary research is necessary to identify relevant blocks, select appropriate tactics, and tailor materials. Through focus groups and the South Carolina Youth Tobacco Survey (SCYTS), specific audiences and relevant blocks in SC were identified, resulting in the three-pronged approach depicted in Figure 1.

#### 1.1.1. Culturally Tailored Campaigns

One element of SC DPH’s program was culturally tailored social media education campaigns for higher-risk teens. These campaigns used peer crowd segmentation—the process of identifying youth subcultures with shared values, norms, and behaviors—to prioritize audiences [38,39,40,41]. In SC, separate campaigns were implemented for the Country and Hip Hop peer crowds based on SCYTS results [42]. The Country peer crowd—described as valuing patriotism and individual liberty and engaging in outdoor activities like hunting and fishing—was particularly at risk for SLT use, while the Hip Hop crowd—described as valuing confidence, a drive to succeed despite obstacles and authenticity—was at risk across products [38,42,43]. Qualitative research indicated that these youth knew tobacco use was harmful, but they discounted risks because the behavior had cultural capital, such as signaling subculture membership. Therefore, the campaigns addressed Impact Blocks by triggering central processing to help youth not ready to change to connect the harms of tobacco use to their values and experiences and addressed Solution Blocks by modeling how teens ready to change could live tobacco-free within their cultural identity [36,37,39,44]. Social media was the primary method of message delivery as it allowed for tailored messages to be delivered efficiently using platforms’ targeting tools as well as organic growth through audience members’ social networks [20,21].

For the Country peer crowd, in 2019, SC DPH and Rescue launched *Down and Dirty* (*D&D*), a social media campaign for youth ages 13–20 focused on SLT, cigarettes, and vaping [39]. *D&D* featured up to two video advertisements and complementary social media posts annually. For the Hip Hop peer crowd, in 2015, the U.S. Food and Drug Administration (FDA) and Rescue launched *Fresh Empire* (*FE*), a tailored youth tobacco education campaign [40]. *FE* sought to educate a national youth audience about the risks of cigarette use through tailored content on social media, television, print, out-of-home, and in-person events across 36 media markets [40,45,46]. SC DPH amplified *FE* within the state by promoting campaign content to SC teens on Instagram and Facebook.

#### 1.1.2. Vaping Education Campaign

The second element of SC DPH’s program, *Behind the Haze* (*BTH*), was a broad social media campaign addressing nicotine vaping among youth ages 13–20. *BTH* was developed through research, which indicated that teens at risk for vaping were in the early stages of change as they knew little about the potential harms and largely discounted the risks they did know [36,47]. Therefore, *BTH* addressed Information Blocks by grabbing the audience’s attention with novel foundational knowledge about the chemical content, addiction risk, and physical and mental health effects of vaping. Additionally, *BTH* addressed Impact Blocks by personalizing risks and showcasing realistic, short-term effects that were harder to ignore than theoretical long-term effects. *BTH* was disseminated through up to three video advertisements and complementary social media posts annually.

#### 1.1.3. Vaping Cessation Program

The third element of SC DPH’s program addressed teens’ desire to quit vaping. Although many young people who vape want to quit, few use existing cessation resources such as quitlines [48,49,50,51,52]. Instead, they want digital programs that are specific to vaping, made for young people, provide tools to reduce stress, and foster social support [33,48,53,54,55]. However, few cessation programs incorporate these preferences, and most deliver one-way text messages to individuals rather than interactive group activities [56,57,58]. To fill this gap, Rescue, Hopelab, and the University of California San Francisco Center for Tobacco Control Research and Education created *Quit the Hit* (*QTH*), an Instagram-based vaping cessation program for youth ages 13–20 [59]. Building on techniques developed in young adult smoking cessation Facebook groups [60,61,62], *QTH* provided 30 days of online support to groups of 10–15 youth with active moderation by a cessation coach. This program was conducted through Instagram group direct messages (DMs), repurposing a communication mode favored by teens. *QTH* assisted youth in the Action and Maintenance stages of the TTM by addressing Solution Blocks, such as struggling to envision their lives without vaping or what quitting looks like, and Reinforcement Blocks, such as lacking social support and healthy coping strategies [36]. The coach used the group DM to send daily educational content covering topics, including predatory marketing of vape products, the pathway from vaping to smoking initiation, signs and symptoms of nicotine addiction, and ways to improve chances of successfully quitting, such as disposing of one’s vapes and practicing alternate behaviors [59]. The coach also used the group DM to set challenges and create opportunities for youth to ask questions, share struggles, set a quit date, and support each other. Content enabled youth to identify their use patterns, taught coping skills for cravings and triggers, and normalized relapse and trying again. In SC, 516 youth ages 13–20 enrolled in 2021–2023.

### 1.2. Study Objectives

This study presents results from a program evaluation of SC DPH’s youth tobacco control efforts on social media. To understand the reach and reception of the education campaigns, we examined youth responses to online, cross-sectional surveys measuring awareness and perceptions of *D&D* and *BTH* plus tobacco knowledge, attitudes, and beliefs. Given the FDA’s robust evaluation of *FE*, we did not track its reach and reception in SC [45,63]. To understand youth perceptions of *QTH* and potential behavioral impact, we examined baseline and follow-up surveys from program participants, with a focus on how youth felt about the program and behavioral changes. Results provide preliminary insight into the impact of SC DPH’s social media program to inform practice in advance of an economic evaluation and a randomized controlled trial of *QTH* [59].

## 2. Materials and Methods

We conducted annual cross-sectional surveys for *D&D* and *BTH* and baseline and follow-up surveys for *QTH*. All data collection was completed online on participants’ own devices and was approved by Advarra IRB (protocol # Pro00016024) and SC DPH IRB (protocol # IRB. 19-008 and IRB. 20-019). Participants were recruited via paid advertisements on Facebook, Instagram, and Snapchat, market research panels, and peer referrals. We also invited cross-sectional survey participants to participate in future waves via email or text and used email and Instagram DM to deliver follow-up surveys to *QTH* participants. Interested youth completed a brief screening survey to determine eligibility: ages 13–20, residents of South Carolina, and vaped at least weekly (*QTH* only). Eligible youth then provided electronic assent (ages 13–17) or consent (ages 18–20) and received a parental opt-out form via email to deliver to their guardian.

### 2.1. BTH and D&D Cross-Sectional Surveys

#### 2.1.1. Data Collection

Cross-sectional survey data were collected in September–October 2019, June–August 2020, September–October 2022, and April–May 2023 for *D&D* (N = 1149), and in November–December 2020, April–May 2022, and February–April 2023 for *BTH* (N = 827). All data were collected after the respective campaigns had launched in SC (*D&D*: March 2019; *BTH*: January 2020). Data were collected separately for the campaigns due to timing and the use of targeting keywords in social media recruitment to increase the likelihood that members of their distinct audiences would see the recruitment advertisements.

#### 2.1.2. Measures

Survey instruments can be found in the Appendix A.

Demographics. We collected data on gender identity (reported as male, female, or another identity) and race/ethnicity (reported in mutually exclusive categories). We calculated age using reported birthdate and rural or urban residence using zip code and census population density (rural: <1000 residents per square mile; urban: ≥1000 residents per square mile).

Tobacco risk. Participants reported their ever use and past 30-day use (dichotomized to yes/no) of nicotine vapes, SLT, cigarettes, CLCCs, and cannabis or marijuana vapes. Participants who had never used nicotine vapes, SLT, or cigarettes were asked four questions per product to assess susceptibility to future use [64,65]. Participants who selected “definitely yes”, “probably yes”, or “probably not” for at least one question were considered susceptible to that product.

Peer crowd identification. For *D&D*, participants completed the I-Base Survey^®^, a photo-based tool assessing peer crowd identification [38,39,45,63,66]. Participants viewed 80 photos of unknown teens representing five peer crowds (Alternative, Country, Hip Hop, Mainstream, Popular) and selected the three male and three female photos that most fit with their friends and three male and three female photos that least fit with their friends. Participants earned positive points for the peer crowds of photos they selected as best fit and negative points for the crowds of photos they selected as least fit, with total scores for each crowd ranging from −12 to 12. Participants with a positive score for Country (1–12) were considered part of that crowd.

Campaign awareness and reception. To assess campaign awareness, participants viewed a list of tobacco education campaigns, including *D&D* and *BTH*, and indicated if they had heard of each before [67,68,69]. Participants who reported awareness were asked to indicate how much they liked the campaign on a scale from 1 to 5 [39,44].

Ad awareness and reception. Participants viewed two campaign video advertisements that had been promoted within the previous year and indicated if they had seen each one before [70]. Those responding “yes” to at least one video were considered to have video awareness. Participants rated the believability of the advertisements’ claims from 1 to 5. In 2019–2022, participants also completed six questions to assess perceived effectiveness (PE), a validated measure of an advertisement’s potential to impact beliefs and behavior [71,72,73]. Participants’ responses to the PE questions were averaged to generate a score from 1 to 5, with a higher score indicating greater potential for positive impact.

Knowledge. Participants were shown a series of factual statements featured in campaign advertisements and asked to indicate if they had heard each one before.

Attitudes and Beliefs. To assess attitudes and beliefs about tobacco use, participants were shown a series of statements and asked to indicate their disagreement/agreement with each one on a scale from 1 to 5 [39,74].

### 2.2. QTH Baseline and Follow-Up Surveys

#### 2.2.1. Data Collection

Survey data were collected from *QTH* program participants in 2021–2023 at baseline before the program, 30 days after program completion (follow-up 1, F1), and at 60–90 days after program completion in 2022–2023 only (follow-up 2, F2). Eligible youth enrolled by creating a profile on the *QTH* website and providing their Instagram handle. Once the program facilitator verified the participant’s handle, they completed the baseline survey and were invited to a cessation group DM. Follow-up surveys were delivered via Instagram DM and email.

#### 2.2.2. Measures

Demographics. At baseline, participants provided demographic information, including their age and gender identity (reported as female, male, another identity, or prefer not to say). They also provided race/ethnicity data, which are reported here in mutually exclusive categories. Similar to the cross-sectional surveys, we used zip codes to determine if the participants lived in a rural or urban area.

Reason for joining. In 2023 only, at baseline participants were asked to select all that applied from a list of reasons why they may have joined *QTH*.

Program completion. At F1, participants were asked how many of the 30 Instagram messages from the group they read [62].

Program feedback. At follow-up, participants indicated how much they agreed with eight statements about *QTH* on a scale from 1 to 5 [61,62]. At F1, participants reported if they thought they would refer back to program content. At F2, participants reported if they had referred back to the content.

Cessation expectations. In each survey, participants reported how successful they expected to be in quitting nicotine vapes on a scale from 1 to 10 [59]. In 2021–2022, participants also reported in each survey how confident they felt in their ability to stay completely nicotine-free in the next six months on a scale from 1 to 5 [59].

Tobacco and marijuana product use. In each survey, participants reported on how many days in the past 30 days they had used nicotine vapes, cigarettes, SLT, CLCCs, and cannabis or marijuana vapes. Responses were also dichotomized to indicate any current use.

### 2.3. Analysis

All analyses were conducted using SPSS Subscription for Mac with listwise deletion for missing data. *QTH* analyses were limited to participants who responded to the baseline survey and at least one follow-up (223 of 516 youth, 43.2%). To describe the samples, we examined proportions for demographics, current use, and susceptibility among non-users, limiting *QTH* data to baseline responses.

For *D&D* and *BTH*, we first examined campaign reach and reception by assessing the proportion of participants reporting any campaign or video awareness and calculating mean campaign likability (1–5) among campaign-aware participants. We also examined mean believability (1–5) and PE (1–5) across all ads. To understand if results differed across participants, we compared rates or means between subgroups using χ^2^ and one-way ANOVA tests with follow-up testing with Bonferroni correction to identify differences between specific subgroups. For *BTH*, we compared current vape users (any 30-day use; 24.3% of the sample), non-current users (ever vaped, no 30-day use; 18.9%), susceptible non-triers (never vaped, susceptible to future use; 32.8%), and non-susceptible non-triers (never vaped, not susceptible; 24.1%). For *D&D*, we compared Country current vape or SLT users (Country score of 1–12, current vape or SLT use; 19.5% of the sample), Country non-users (Country score of 1–12, no current use; 32.3%), non-Country current users (Country score of −12 to 0, current vape or SLT use; 18.5%), and non-Country non-users (Country score of −12 to 0, no current use; 29.8%). Finally, we examined key outcomes by comparing the proportion of campaign-aware and unaware participants who recalled each campaign fact using χ^2^ tests and comparing mean attitude and belief responses using two-sided *t*-tests.

For *QTH*, we first assessed program perceptions by examining the proportion of participants who selected each reason for joining *QTH*, viewed at least half of the Instagram posts, agreed with program feedback statements, and indicated that they would or already had referred back to program content. To assess potential impact, we examined changes in beliefs and behaviors across surveys. We used one-way ANOVAs to compare mean scores for expectation of success and confidence in staying vape-free between baseline, F1, and F2. For behavior, we used χ^2^ tests to compare the proportion of participants who reported any current use at baseline, F1, and F2 for nicotine vapes, SLT, cigarettes, CLCCs, and marijuana vapes, with follow-up testing with Bonferroni correction to identify specific differences between survey waves. We also calculated the change in the number of days using each product by subtracting a participant’s baseline response from their F1 and F2 responses (range: −30 to 30). To assess if the mean number of days changed significantly, we conducted two-sided *t*-tests comparing the mean difference at F1 and F2 to 0.

## 3. Results

The majority of participants lived in urban zip codes (65.0–70.7%), identified as female (56.1–64.8%), and were recruited via paid social media advertisements (59.0–81.4%) (Table 1). In the *BTH* sample, 24.3% currently vaped nicotine, and 57.7% of non-users were susceptible. In the *D&D* sample, 5.9% currently used SLT, and 20.8% of SLT non-users were susceptible, while 36.6% currently vaped nicotine, and 35.7% of vape non-users were susceptible.

### 3.1. D&D and BTH

Thirty-seven percent of participants recalled each campaign (Table 2). For *D&D*, a significantly greater proportion of Country users recalled the campaign than all other groups (51.8%, *p* < 0.05). On average, the appeal was neutral for *BTH* (*M* = 2.86) and positive for *D&D* (*M* = 3.70), with Country users liking *D&D* more (*M* = 3.89) than non-Country non-users (*M* = 3.53, *p* < 0.05). Campaign awareness and appeal for *BTH* did not vary significantly across subgroups.

For both campaigns, approximately 40.0% of participants recalled at least one video advertisement, with recall being significantly higher among Country users (55.4%) and non-users (44.5%) than non-Country non-users (32.7%, *p* < 0.05). Participants found the ads believable (*BTH M* = 4.21, range: 4.11–4.28; *D&D M* = 4.17, range: 4.00–4.25), with lower believability ratings reported by vape current users (3.91) than all others in the *BTH* sample (4.16–4.42) and by non-Country users (4.00) than Country non-users (4.26) in the *D&D* sample (all *p* < 0.05). *D&D* ads received an average PE score of 3.74 (range: 3.56–3.88), with Country non-users rating the ads higher than all others (3.91 vs. 3.53–3.71, *p* < 0.05). *BTH* ads received an average PE score of 3.76 (range: 3.69–3.98), with non-susceptible non-triers scoring the ads higher than current and non-current vapers (4.01 vs. 3.43–3.73, *p* < 0.05).

Recall of facts from campaign advertisements was significantly higher among campaign-aware than unaware participants for nine out of 10 *D&D* facts and five out of nine *BTH* facts (Table 3). The largest differences between campaign-aware and unaware participants were for *D&D* facts about financial costs and the impact of SLT use on siblings and *BTH* facts about formaldehyde and immunity. For *D&D*, campaign-aware participants reported lower agreement with the statement, “Chewing tobacco is a waste of money” (*p* = 0.016). Attitudes and beliefs did not differ significantly by campaign awareness for *BTH*.

### 3.2. QTH

Among baseline participants who completed at least one follow-up survey, the most common reasons for joining *QTH* were that the program provided expert support (67.3%) and the ability to talk to peers who also wanted to quit (63.3%) (Table 4). Reasons for joining did not differ significantly between participants who did and did not complete at least one follow-up survey (results not shown). Most read more than half of the Instagram messages (91.6%) and would recommend the program to others (90.7%). At F1, 86.4% planned to refer back to the program’s content; at F2, 76.1% had referred back to it.

Beliefs and behaviors differed significantly from baseline to follow-up (Table 5). Compared to baseline, at both follow-ups, participants reported significantly greater expectations that they would succeed in quitting (6.33 vs. 8.11, 7.81, *p* < 0.05) and confidence that they could stay nicotine-free (2.89 vs. 3.74, 4.00, *p* < 0.05). The proportion of participants who vaped nicotine in the prior 30 days decreased significantly from baseline (99.6%) to F1 (80.4%) and from F1 to F2 (50.4%, both *p* < 0.05). Participants vaped nicotine on 19 fewer days at F1 and 23 fewer days at F2 than baseline, both of which differed significantly from 0 (both *p* < 0.001). Participants also reported statistically significant declines in cigarette, CLCC, and marijuana vape use.

## 4. Discussion

In this study, we reported program evaluation results for three elements of SC DPH’s youth tobacco control program: the *D&D* culturally tailored social media campaign, the *BTH* vape education social media campaign, and the *QTH* vaping cessation program. The social media campaigns effectively reached their target audiences with content that was well-received and may have increased knowledge. *QTH* engaged participants and may have contributed to reduced vaping and other tobacco use. The successes of these initiatives and the comprehensive design of SC DPH’s youth tobacco control program provide a template for other states seeking to establish an equitable response to youth tobacco use on social media.

Few evaluations of state-level youth tobacco education campaigns delivered via social media have been published, leaving some questions as to what constitutes success for local efforts. Over one-third of participants recalled SC DPH’s social media education campaigns, and nearly half recalled at least one video advertisement, which aligns with the limited state-level data available [75]. While national campaigns have achieved awareness rates above 70%, those campaigns had larger budgets and employed both traditional media channels such as television and social media [45,63,67,68,69,76,77]. Further research is needed to establish awareness benchmarks for local social media education campaigns, but given the link between televised advertisements and campaign awareness [78], achieving half or more of national campaigns’ awareness rates using social media alone may indicate its promise for reaching teens. Additionally, for the culturally tailored *D&D* campaign, which was designed to appeal to a specific subculture of at-risk teens, awareness was higher among target audience members than other participants. This indicates that tailoring content, style, and delivery platforms, as was performed for *D&D*, may increase a social media campaign’s efficiency in reaching its target audience compared to a more generalized approach [44]. While awareness provides evidence of the campaigns’ reach, further research is needed to understand the role of campaign engagement, not only awareness, in behavior change and how different types and intensities of social media engagement may contribute to individual change [79].

*BTH* and *D&D* were well-received, with advertisement PE scores in line with those reported for successful national youth tobacco campaigns such as *The Real Cost* (3.40–4.07) and *FE* (3.53–4.11) [45,63,68,80,81]. Several studies have linked advertisement PE scores to behavioral intentions and behavior change, particularly for tobacco use [71,72], indicating the advertisements had the potential to positively impact SC teens. Further, recall of many facts featured in campaign advertisements was higher among campaign-aware participants than campaign-unaware participants, indicating that the campaigns may have increased knowledge about the harms of tobacco use in line with previous systematic reviews and meta-analyses of the effects of social media health education [23,82]. In particular, we believe that the campaigns may have succeeded in cutting through the glut of health information and misinformation on social media to increase knowledge because message content was carefully selected based on formative research with the campaign audiences to identify their stages of change, existing knowledge, and barriers to change. At the same time, agreement with most attitude and belief statements did not differ significantly between campaign-aware and unaware participants. This could be due to a ceiling effect, as many statements had high rates of agreement across groups. However, prior evaluations have also found mixed evidence linking campaign awareness to changes in targeted beliefs, indicating that further research is needed to understand the role attitudes and beliefs may play in the causal pathway from social media campaign exposure to intentions and behavior [63,83,84,85,86,87]. A future economic evaluation of SC DPH’s program may provide insight into whether these observed effects translate to population-level changes in youth tobacco use, a much-needed contribution to the field of social media health communications [22,24].

*QTH* participants reported strong engagement and satisfaction with the program. Not only did most report the program was easy to understand, helpful, actionable, and worth recommending, but also three-quarters indicated they had referred back to program content after completion. Results match or surpass those reported for the Tobacco Status Project, a similar young adult smoking cessation program facilitated via Facebook groups [61]. Positive reception may be due to *QTH*’s unique features—a moderated, interactive, social media-based support group of peers focused on vaping cessation—which align with youth preferences for vape cessation programs [33,88].

Preliminary evidence presented here also indicates that *QTH* positively impacted behavior, as the proportion of participants who vaped any nicotine and the average number of days on which they vaped decreased significantly from baseline to follow-up. This aligns with results from a randomized controlled trial of Tobacco Status Project cessation groups, which observed a positive effect on smoking cessation at three-month follow-up, though no effects were observed at 12 months [61]. Interestingly, the use of cigarettes, CLCCs, and marijuana vapes also decreased significantly among *QTH* participants. This halo effect has been observed in other studies and should be explored further [89]. We believe these positive effects can be attributed especially to the interactive nature of the program, which leveraged the “social” aspect of social media to generate peer support and engagement in the group DMs. This is a unique benefit of *QTH*’s approach over mass media or one-way communication interventions such as text-messaging programs, which will be further assessed in a future randomized controlled trial [59].

Taken together, findings from the current study can inform the development of state-level youth tobacco control efforts. Specifically, SC DPH’s decision to adopt a multi-pronged social media approach appears to have succeeded by reaching at-risk teens across the state, possibly increasing their knowledge, and potentially enabling youth to quit vaping. We believe practitioners can learn from SC DPH’s approach by identifying at-risk populations in their communities and selecting tactics and media channels to promote behavior changes that are designed specifically for the stages of change and barriers those audiences face. States seeking to modernize their approach to youth tobacco control should also aim to leverage the latest social media developments to reach and engage teens effectively, as achieved in SC DPH’s program, particularly *QTH*. Tailored tobacco control efforts delivered via social media can enable localities to maximize the reach and impact of available funding for youth prevention and cessation.

### Limitations

For the education campaign cross-sectional surveys, we were unable to conduct longitudinal analyses due to the limited sample size of follow-up participants, precluding assessments of causality or temporality. The *QTH* surveys lacked randomization, a control group, and biochemical validation of nicotine abstinence; these limitations will be addressed in a future randomized controlled trial [59]. Additionally, only 43.2% of *QTH* program participants were included in analyses due to drop-off between baseline and follow-up surveys, which may bias results. For all three studies, samples were nonrepresentative, which may limit generalizability and preclude us from drawing conclusions about the population-level effects of the program. Finally, because each data collection was conducted independently, we are unable to draw conclusions about the potential additive or synergistic effects of the campaigns on youth tobacco use in SC.

## 5. Conclusions

In this study, we described early insights into the reach, reception, and impact of two youth tobacco education social media campaigns and a novel youth vaping cessation program delivered entirely via Instagram. While each program element contributed independently to SC DPH’s goals, the combination of broad education, tailored education, and a cessation program is what distinguishes SC DPH’s approach to social media health education. State youth tobacco control programs have historically centered around a singular generalized education campaign that attempts to reach all teens equally, often via mass media channels such as television. However, tobacco use is not equally distributed among teens, and those at greatest risk are often those most difficult to reach efficiently using standard techniques [21,23]. The result is that funding is disproportionately spent reaching lower-risk youth, which furthers inequity. While having multiple carefully crafted social media campaigns may appear less efficient, in fact, it allows resources to be funneled directly to populations facing tobacco use disparities, increasing both equity and efficiency [90]. Further, the approach described herein was facilitated by the advent of social media, which has fractured the traditional media environment in a way that enables tailored messaging to be delivered on platforms carefully selected to align with the target audience’s preferences, further reducing waste [21,22,90]. Social media encourages innovation, with promising programs such as *QTH* leveraging teens’ existing social media activities to promote behavior change through the social aspects of social media. Future economic evaluations and a randomized trial of the *QTH* program will provide additional insight into the cost-effectiveness and population-level impact of SC DPH’s social media youth tobacco control program.

## Figures and Tables

**Figure 1 ijerph-22-00269-f001:**
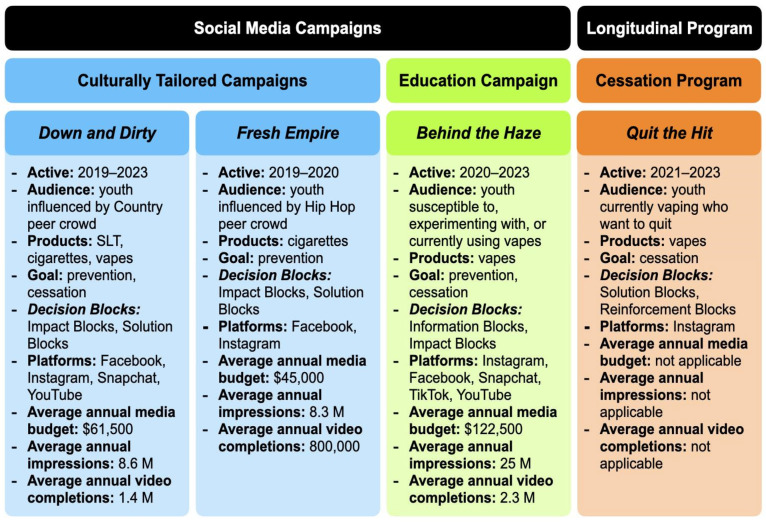
Three elements of SC DPH’s youth tobacco control program, 2019–2023. Notes: SC DPH, South Carolina Department of Public Health; SLT, smokeless tobacco.

**Table 1 ijerph-22-00269-t001:** Sample descriptives.

Category	*D&D* (N = 1149)	*BTH* (N = 827)	*QTH* (N = 223)
Year, % (n)			
2019	33.0 (379)	---	---
2020	27.4 (315)	47.2 (390)	---
2021	---	---	34.1 (76)
2022	19.2 (221)	29.5 (244)	22.0 (49)
2023	20.4 (234)	23.3 (193)	43.9 (98)
Geography, % (n)			
Urban	69.3 (796)	70.7 (585)	65.0 (145)
Rural	30.7 (353)	29.3 (242)	35.0 (78)
Age, % (n)			
13–17	37.3 (429)	61.8 (511)	58.7 (131)
18–20	62.7 (720)	38.2 (316)	41.3 (92)
Gender identity, % (n)			
Female	62.1 (714)	64.8 (536)	56.1 (125)
Male	35.4 (407)	33.7 (279)	35.0 (78)
Another identity	2.4 (28)	1.5 (12)	7.6 (17)
Prefer not to say	---	---	1.3 (3)
Race/ethnicity, % (n)			
Hispanic	6.6 (76)	9.9 (82)	11.2 (25)
Non-Hispanic White	74.8 (859)	63.0 (521)	57.4 (128)
Non-Hispanic Black	9.5 (109)	16.1 (133)	18.4 (41)
Non-Hispanic Asian, Native Hawaiian, or Pacific Islander	1.3 (15)	2.5 (21)	1.3 (3)
Non-Hispanic multiracial	5.2 (60)	7.0 (58)	8.1 (18)
Non-Hispanic another identity	2.6 (30)	1.5 (12)	0.9 (2)
Prefer not to say	---	---	2.7 (6)
County peer crowd, % (n)	51.8 (595)	---	---
Recruitment method, % (n)			
Social media	81.4 (935)	59.0 (488)	---
Panel	14.2 (163)	26.4 (218)	---
Follow-up	4.4 (51)	14.6 (121)	---
Current tobacco use, % (n)			
Vape	36.6 (421)	24.3 (201)	99.6 (222)
SLT	5.9 (68)	1.5 (12)	4.0 (9)
Cigarettes	15.1 (174)	9.4 (78)	33.6 (75)
CLCCs	12.0 (138)	5.3 (44)	20.6 (46)
Marijuana vape ^1^	17.1 (132)	17.3 (143)	51.6 (115)
Susceptible non-users, % (n) ^2^			
Vape	35.7 (169)	57.7 (271)	---
SLT	20.8 (203)	---	---
Cigarettes	46.1 (334)	---	---

**Notes:** *BTH*, *Behind the Haze*; CLCCs, cigars, little cigars, and cigarillos; *D&D*, *Down and Dirty*; *QTH*, *Quit the Hit*; SLT, smokeless tobacco. ^1^ For the *D&D* sample, only asked in 2020–2023 (n = 770). ^2^ Among participants in the *D&D* sample reporting they had never used vapes (n = 474), SLT (n = 976), or cigarettes (n = 725), and in the *BTH* sample reporting they had never used vapes (n = 470).

**Table 2 ijerph-22-00269-t002:** *D&D* and *BTH* campaign and video ad awareness and reception.

	*D&D*		*BTH*
Campaign awareness, %	(n = 1149)		(n = 827)
Overall	37.1	Overall	37.2
Audience	***p* < 0.001**	Audience	*p* = 0.225
Country user	51.8 ^a^	Vape current user	43.3
Country non-user	38.3 ^b^	Vape non-current user	36.5
Non-Country user	37.7 ^b^	Vape SNT	35.4
Non-Country non-user	25.7 ^c^	Vape NSNT	34.2
Campaign appeal (1–5), *M* (*SD*)	(n = 426)		(n = 214)
Overall	3.70 (0.95)	Overall	2.86 (1.13)
Audience	***p* = 0.040**	Audience	*p* = 0.171
Country user	3.89 (0.92) ^a^	Vape current user	3.05 (1.09)
Country non-user	3.63 (0.95) ^a,b^	Vape non-current user	2.80 (1.05)
Non-Country user	3.75 (1.01) ^a,b^	Vape SNT	2.91 (1.07)
Non-Country non-user	3.53 (0.90) ^b^	Vape NSNT	2.56 (1.30)
Video ad recall, %	(n = 1149)		(n = 827)
Overall	42.5	Overall	40.7
Audience	***p* < 0.001**	Audience	*p* = 0.445
Country user	55.4 ^a^	Vape current user	38.8
Country non-user	44.5 ^a,b^	Vape non-current user	43.6
Non-Country user	41.0 ^b,c^	Vape SNT	38.0
Non-Country non-user	32.7 ^c^	Vape NSNT	44.2
Video ad believability (1–5), *M* (*SD*)	(n = 1149)		(n = 827)
Overall	4.17 (0.92)	Overall	4.21 (0.87)
Audience	***p* = 0.013**	Audience	***p* < 0.001**
Country user	4.16 (0.96) ^a,b^	Vape current user	3.91 (1.03) ^a^
Country non-user	4.26 (0.87) ^a^	Vape non-current user	4.16 (0.82) ^b^
Non-Country user	4.00 (1.00) ^b^	Vape SNT	4.31 (0.73) ^b,c^
Non-Country non-user	4.18 (0.88) ^a,b^	Vape NSNT	4.42 (0.79) ^c^
Video ad PE score, *M* (*SD*)	(n = 915)		(n = 634)
Overall	3.74 (0.85)	Overall	3.76 (0.84)
Audience	***p* < 0.001**	Audience	***p* < 0.001**
Country user	3.65 (0.91) ^a^	Vape current user	3.43 (1.00) ^a^
Country non-user	3.91 (0.79) ^b^	Vape non-current user	3.73 (0.85) ^b^
Non-Country user	3.53 (0.89) ^a^	Vape SNT	3.83 (0.66) ^b,c^
Non-Country non-user	3.71 (0.81) ^a^	Vape NSNT	4.01 (0.79) ^c^

**Notes**: *BTH*, *Behind the Haze*; *D&D*, *Down and Dirty*; NSNT, non-susceptible non-trier; PE, perceived effectiveness; SNT, susceptible non-trier. Boldface indicates statistical significance. Superscript letters indicate a statistically significant difference (*p* < 0.05) between two audience categories in follow-up comparisons using Bonferroni correction. For a particular measure, two audience categories with different letters differ significantly from each other, while two audience categories with matching letters do not differ significantly from each other.

**Table 3 ijerph-22-00269-t003:** *D&D* and *BTH* knowledge, attitudes, and beliefs among campaign-aware and unaware participants.

*D&D*	Overall	Aware	Unaware	*p*-Value
Knowledge, %				
Dipping and chewing tobacco leads to tooth loss and gum disease (n = 770).	88.7	92.1	86.5	**0.015**
If you use tobacco, your younger siblings or relatives are more likely to also use tobacco (n = 770).	69.6	81.0	62.2	**<0.001**
1300 Americans die each day from tobacco use (n = 536).	52.6	61.9	47.6	**0.001**
Vape aerosol contains up to 31 chemicals like nickel, lead, and benzene that can damage your lungs (n = 536).	74.1	83.6	68.9	**<0.001**
Dip and chewing tobacco can cause mouth cancer (n = 536).	86.2	90.5	83.9	**0.034**
If you quit chewing tobacco, you can save about $1000 a year (n = 536).	53.4	66.1	46.4	**<0.001**
Nicotine changes your brain in a way that affects your mood, making you not feel right (n = 455).	83.3	86.1	81.5	0.193
Chemicals in chew and dip are absorbed through your mouth, which can cause cancer (n = 455).	80.0	90.0	73.5	**<0.001**
Teens who use vapes can spend over $700 in six months on one pack of pods or three disposables per week, which is about $1500 a year (n = 455).	51.2	64.4	42.5	**<0.001**
Vape aerosol contains toxic chemicals like arsenic, lead, and formaldehyde, which, even in small amounts, can cause brain damage, lung disease, or cancer (n = 455).	80.2	88.9	74.5	**<0.001**
Attitudes and beliefs (1–5), *M* (*SD*)				
Chewing tobacco is dangerous to a person’s health (n = 1149).	4.44 (0.84)	4.42 (0.87)	4.46 (0.83)	0.368
Chewing tobacco is a waste of money (n = 1149).	4.39 (0.93)	4.30 (1.01)	4.44 (0.87)	**0.016**
Using vapes and e-cigarettes is dangerous to a person’s health (n = 1149).	4.22 (1.00)	4.23 (1.02)	4.20 (0.99)	0.624
It is important to me to live a tobacco-free lifestyle (n = 1149).	4.11 (1.11)	4.03 (1.16)	4.15 (1.08)	0.087
** *BTH* **	**Overall**	**Aware**	**Unaware**	***p*-value**
Knowledge, %				
The chemicals in vapes break down the defenses in your lungs, making you more vulnerable to viruses (n = 827).	69.5	77.6	64.7	**<0.001**
Even if you are strong, young, and healthy, the chemicals in vapes can weaken you and damage your lungs on a cellular level (n = 827).	76.2	78.6	74.8	0.213
Vape aerosols contain lead, a neurotoxin that can cause brain damage (n = 827).	61.4	68.8	57.0	**<0.001**
Vape juice and vapor contain dangerous chemicals, like formaldehyde. Formaldehyde is used to preserve dead bodies (n = 827).	60.5	70.1	54.7	**<0.001**
Vaping can cause a chemical burn in the lungs, similar to that seen in people exposed to poisons like mustard gas (n = 634).	56.3	61.2	53.8	0.075
Vape companies went into schools and lied to teens, telling them that vapes are “totally safe” (n = 634).	53.6	60.7	50.0	**0.010**
Nicotine can mess with neurotransmitters in the brain that are linked to stress, anxiety, and depression (n = 437).	78.0	80.6	76.3	0.295
Nicotine affects the natural balance of neurotransmitters in the brain, such as dopamine and serotonin, which are important for stabilizing mood (n = 437).	65.7	69.1	63.4	0.212
Sharing vapes with friends also means sharing nicotine addiction (n = 437).	53.8	60.6	49.2	**0.020**
Attitudes and beliefs (1–5), *M* (*SD*)				
If I were to use a vape, I would worry about my health risks (n = 827).	4.03 (1.35)	3.94 (1.40)	4.08 (1.33)	0.143
If I were to use a vape, I would harm my lungs (n = 827).	4.10 (1.33)	4.12 (1.29)	4.08 (1.35)	0.711
If I were to use a vape, I would worry about the chemicals that I am inhaling (n = 827).	3.87 (1.44)	3.79 (1.48)	3.92 (1.41)	0.192
If I were to use a vape, I would put myself at risk for addiction (n = 827).	3.24 (1.59)	3.28 (1.57)	3.22 (1.60)	0.571
If I were to use a vape, I would worry about how it would affect my brain (n = 437).	3.76 (1.42)	3.74 (1.41)	3.78 (1.43)	0.765
If I were to use a vape, I would worry about how it would affect my emotions (n = 437).	3.59 (1.45)	3.54 (1.49)	3.62 (1.42)	0.566

**Notes**: *BTH*, *Behind the Haze*; *D&D*, *Down and Dirty*. Boldface indicates statistical significance (*p* < 0.05).

**Table 4 ijerph-22-00269-t004:** *Quit the Hit* program feedback.

	Baseline (n = 98)	F1 (n = 214)	F2 (n = 113)
Reason for joining, % ^1^			
It provides expert support	67.3	---	---
I can talk to my peers/people who also want to quit	63.3	---	---
I can get money for taking the surveys	58.2	---	---
I can access the group online	42.9	---	---
Another reason	5.1	---	---
Participation, %			
Read more than half of the Instagram messages from the group	---	91.6	---
Program feedback, %			
I believe the facilitator gave sound advice	---	92.1	---
I would recommend this program to others	---	90.7	---
The messages from the facilitator were easy to understand	---	90.2	---
The messages from the facilitator gave me something new to think about	---	88.3	---
I have thought about what I read in the group	---	89.7	87.6
The messages from the facilitator have helped me to be healthier	---	89.7	82.3
I have used information shared in the group	---	87.9	89.4
I tried to take action on the suggestions/quit tips shared in the group	---	87.3	85.8
Content feedback, %			
I will refer back to the content posted in the Instagram group for more information and ideas for changing my vaping	---	86.4	---
I referred back to the content posted in the Instagram group for more information and ideas for changing my vaping	---	---	76.1

**Notes**: F1, follow-up 1; F2, follow-up 2. ^1^ Only asked in the 2023 baseline survey.

**Table 5 ijerph-22-00269-t005:** Beliefs and 30-day tobacco use at baseline and follow-up among *Quit the Hit* participants.

	Baseline (n = 223)	F1 (n = 214)	F2 (n = 113)	*p*-Value
Beliefs, *M* (*SD*)				
How successful participant expects to be in quitting vaping (1–10)	6.33 (2.25) ^a^	8.11 (1.87) ^b^	7.81 (2.28) ^b^	**<0.001**
Confidence in staying completely nicotine-free in next six months (1–5) ^1^	2.89 (1.16) ^a^	3.74 (1.14) ^b^	4.00 (1.06) ^b^	**<0.001**
Any 30-day use, %				
Nicotine vapes	99.6 ^a^	80.4 ^b^	50.4 ^c^	**<0.001**
SLT	4.0	2.3	4.4	0.510
Cigarettes	33.6 ^a^	21.0 ^b^	17.7 ^b^	**0.001**
CLCCs	20.6 ^a^	8.9 ^b^	7.1 ^b^	**<0.001**
Marijuana vapes	51.6 ^a^	40.2 ^a,b^	34.5 ^b^	**0.005**
	**F1-Baseline** (n = 214)	***p*-value**	**F2-Baseline** (n = 113)	***p*-Value**
Difference in number of days used in past 30 (−30–30), *M* (*SD*)				
Nicotine vapes	−19.44 (9.68)	**<0.001**	−22.72 (9.14)	**<0.001**
SLT	−0.23 (3.86)	0.377	−0.18 (4.47)	0.675
Cigarettes	−1.30 (7.17)	**0.009**	−1.01 (5.64)	0.060
CLCCs	−1.12 (5.53)	**0.004**	−0.68 (5.91)	0.223
Marijuana vapes	−2.75 (10.00)	**<0.001**	−3.06 (9.58)	**<0.001**

**Notes**: F1, follow-up 1; F2, follow-up 2; CLCCs, cigars, little cigars, and cigarillos; SLT, smokeless tobacco. Boldface indicates statistical significance (*p* < 0.05). Superscript letters indicate statistically significant difference (*p* < 0.05) between two survey waves in follow-up comparisons using Bonferroni correction. For a particular measure, two survey waves with matching letters do not differ significantly from each other. ^1^ Only asked in 2021 and 2022.

## Data Availability

The data underlying this study are not publicly available due to confidentiality agreements with participants. However, reasonable requests for data may be made to the corresponding author, and these will be considered on a case-by-case basis in compliance with ethical and legal restrictions.

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
