# Peer review of "Youth Tobacco Control in the Digital Age: Impact of South Carolina’s Youth Tobacco Education and Vaping Cessation Social Media Programs"

_ijerph, 2025, doi:10.3390/ijerph22020269_

Round 1

Reviewer 1 Report

Comments and Suggestions for Authors

Authors could have used better visualization to present their story.

Reviewer 2 Report

Comments and Suggestions for Authors

This study on the effects of tobacco-use and vaping cessation campaigns among young people in South Carolina is one of the most well written, most solid scientific papers I have read in my entire career as a peer-reviewer. I only have a small rather stylistic suggestion on page 5 (segment 1.2.). Please see attached file.

However, I have a huge concern: conflict of interest between the initiators and funders of the campaigns themselves described and analyzed in this paper and the funders of the research itself which led to this paper: they are the same, SCDHEC, with all authors being employed either by SCDHEC or by the partner institution in this program Rescue Agency. See higghlighted segments on pages 1 and 15.  In the segment related to conflicts of interest on page 15, this appears clearly stated. To me, as an European person living and working in an Asian space, the fact that employees of institutions which have developed social programs pursue and analyze the effects of the very programs their employers had developed is absolutely a conflict of interest as it stands to logic that this is a conflict of interest: by no means could the employees ever be objective in observing, analyzing and evaluating their own employers' programs as they themselves are part of those employers' ecosystems. Moreover, a positive evaluation of such programs is meant and usually leads to a continuation of the funding itself, with all the legal, professional and socioeconomic implications.

Maybe in the US-American system this type of structural interactions is normal - as lobbying is (which, again, from a European perspective is pure bribing to the benefits of those in power with socioeconomic and political power and the detriment of "the rest"). From a non-US-American perspective, this is clearly a conflict of interest problem which ought to be addressed BEFORE the publication which will be open to global readership.

I also do not have a solution to this situation: perhaps involve some experts who are NOT hired or in any sort of contractual arrangement with either SCDHEC or Rescue Agency and who could deliver the external validation to this research. As mentioned at the beginning of these lines, the research leading up to the paper seems solid and objective, but the entire framework appears as highly controversial due to the employee-employer relationship of apparently ALL authors of the paper with either one of the major institutions involved in the health campaigns themselves.

Author Response

  1. Summary

We would like to thank the reviewer for taking the time to provide feedback on our manuscript. Please find our detailed response to specific comments below.

  1. Questions for General Evaluation

Questions for general evaluation

Reviewer’s evaluation

Response and revisions

Does the introduction provide sufficient background and include all relevant references?

Yes

We would like to thank the reviewer for their kind response, and hope that we have addressed their specific feedback below.

Is the research design appropriate?

Yes

Are the methods adequately described?

Yes

Are the results clearly presented?

Yes

Are the conclusions supported by the results?

Yes

  1. Point-by-point response to Comments and Suggestions for Authors

Comment 1: This study on the effects of tobacco-use and vaping cessation campaigns among young people in South Carolina is one of the most well written, most solid scientific papers I have read in my entire career as a peer-reviewer. I only have a small rather stylistic suggestion on page 5 (segment 1.2.). Please see attached file.

Response 1: We would like to thank the reviewer for taking the time to provide feedback on our manuscript. We have made this stylistic edit (replaced “manuscript” with “study” in the first sentence of section 1.2, Study Objectives, page 5, line 182).

Comment 2: However, I have a huge concern: conflict of interest between the initiators and funders of the campaigns themselves described and analyzed in this paper and the funders of the research itself which led to this paper: they are the same, SCDHEC, with all authors being employed either by SCDHEC or by the partner institution in this program Rescue Agency. See highlighted segments on pages 1 and 15.  In the segment related to conflicts of interest on page 15, this appears clearly stated. To me, as an European person living and working in an Asian space, the fact that employees of institutions which have developed social programs pursue and analyze the effects of the very programs their employers had developed is absolutely a conflict of interest as it stands to logic that this is a conflict of interest: by no means could the employees ever be objective in observing, analyzing and evaluating their own employers' programs as they themselves are part of those employers' ecosystems. Moreover, a positive evaluation of such programs is meant and usually leads to a continuation of the funding itself, with all the legal, professional and socioeconomic implications.

Maybe in the US-American system this type of structural interactions is normal - as lobbying is (which, again, from a European perspective is pure bribing to the benefits of those in power with socioeconomic and political power and the detriment of "the rest"). From a non-US-American perspective, this is clearly a conflict of interest problem which ought to be addressed BEFORE the publication w hich will be open to global readership.

I also do not have a solution to this situation: perhaps involve some experts who are NOT hired or in any sort of contractual arrangement with either SCDHEC or Rescue Agency and who could deliver the external validation to this research. As mentioned at the beginning of these lines, the research leading up to the paper seems solid and objective, but the entire framework appears as highly controversial due to the employee-employer relationship of apparently ALL authors of the paper with either one of the major institutions involved in the health campaigns themselves.

Response 2: We would like to acknowledge the reviewer’s concern regarding potential conflict of interest for this manuscript. Unfortunately for public programs in the United States, funding for independent external evaluations is often very limited, especially for programs funded and implemented at the state or local level such as the one described in this manuscript. Evaluation funding often must be carved out from already-tight program implementation budgets, causing program administrators to make difficult choices regarding how much funding to divert from program implementation (thereby potentially reducing program reach and impact) into evaluation.

In the case of the tobacco control program described in the current study, SCDHEC elected to set aside funding to conduct both a program evaluation (described in this manuscript) and an economic evaluation. The purpose of the program evaluation described in this manuscript was to provide timely insight into reach, reception, and potential effects of the program. It is important to note that while Rescue Agency and SCDHEC were involved in the implementation and funding of the program, the evaluation was conducted with a clear distinction between program management and program evaluation staff. As evaluators, we prioritize rigorous methodologies and unbiased reporting to ensure the integrity of the findings. This program evaluation enabled SCDHEC and Rescue to gather early insight into program performance in a timeframe that allowed for the program to be fine-tuned based on learnings from the evaluation. We approached this program evaluation process with a commitment to rigor, transparency, and objectivity. We have tried to present the available data as fairly as possible, for example including metrics such as attitudes and beliefs for which we did not find positive effects of the social media campaigns (Table 3, pages 10-11), and to be forthright about the authors’ potential conflict of interest as well as the general limitations of the data (see Conflict of Interest statement, page 15, lines 538-542 and Limitations section, page 14, lines 478-488). We hope that dissemination of these program evaluation results can provide learnings for practitioners seeking to implement social media campaigns of their own, and ideas of how implementation of those campaigns can be monitored in real-time to provide insight that can be quickly applied to improve program performance.

Separate from the current study, SCDHEC contracted researchers  from the University of South Carolina and University of Georgia to conduct an economic evaluation of the social media campaigns described in this manuscript (Down and Dirty, Fresh Empire, Behind the Haze). The purpose of this economic evaluation, which is currently ongoing, is to use data from the statewide, representative South Carolina Youth Tobacco Survey to examine population level changes in tobacco related beliefs and behaviors resulting from the program, and cost to the state for the program. We have included a copy of the preliminary report resulting from this effort, which identified increases over time in the proportion of youth reporting exposure to the social media campaigns as well as decreases in several tobacco use measures (though also decreases in the proportion of youth desiring to quit tobacco and who had quit tobacco in the prior 12 months). While this analysis is not yet complete and has a different purpose than the program evaluation reported in the current manuscript, both studies generally align in finding at least some evidence of positive impact of the campaigns.

Additionally, a separate randomized controlled trial is underway to examine the effectiveness of Quit the Hit which will provide clearer evidence on the cessation program’s effects (see https://pubmed.ncbi.nlm.nih.gov/36496358/, also cited in the manuscript). This trial is mentioned several times in the manuscript as a forthcoming source of insight into the effectiveness of QTH (page 13, lines 474-476; page 14, lines 480-482).

As such, we consider our results to be preliminary insights into how the program performed in South Carolina, with further, more rigorous investigations forthcoming from other parties. In the interest of sharing knowledge in a timely manner, we feel dissemination of these results, with proper identification of the conflict of interest, in advance of the economic evaluation and randomized trial can be helpful for practitioners making planning decisions. We hope that this has been reflected throughout the manuscript in the cautious language we have used when interpreting results, the limitations we have noted (including additional language added around our inability to draw population-level impact conclusions, page 14, lines 485-486), and the revisions we have made to section 1.2, Study Objectives (page 5, lines 182-192) and section 5, Conclusions (page 14, lines 512-513). Combined with the conflict of interest statement, we hope that this information assuages the reviewer and editors’ concerns as sufficiently as possible within the constraints of the existing data. We hope this explanation reassures the reviewer of our commitment to transparency, rigor, and the separation of evaluation from program implementation. Disseminating these findings now—with appropriate caveats—can provide timely insights for public health practitioners while laying the groundwork for future, more definitive analyses.

  1. Response to Comments on the Quality of English Language

Response 1: We have made no major changes to the quality of the English language, based on the reviewer’s selection of “The quality of English does not limit my understanding of the research.”

Reviewer 3 Report

Comments and Suggestions for Authors

The manuscript entitled “Youth tobacco control in the digital age: Impact of South Carolina’s youth tobacco education and vaping cessation social media programs” was interesting. However, the following issues need further attentions:

1-     I think the abstract should be structured according to the journal instructions.

2-     Please make the aim of the research clear in the abstract.

3-     The keywords can be selected based on the MeSH Terms.

4-     I think it is better, if the authors merge all subsections of the introduction section and make it shorter. In my opinion, Table 1 is adequate to describe the program and the other 3 subsections can be summarized just in one paragraph.

5-     The number of the participants in each survey should be mentioned in the methods section.

6-     The descriptions of questionnaires are a bit long. I suggest the authors to summarize data collection instruments and provide a copy of the questionnaires as supplementary files.

7-     Please use “sex” instead of “gender”.

8-     Please add “implications for policy” and “implications for practice”.

9-     Please remove references from the conclusion sections. The conclusion section should provide an overview of the findings, the researchers’ reflections on the results, and recommendations for future research.

Round 2

Reviewer 3 Report

Comments and Suggestions for Authors

I appreciate the authors for their time and efforts to revise the manuscript. However, some issues still need further attention.

1-      The main headings of the structured abstract (background, methods, results, conclusions) are missing.

2-      The introduction section and its subsections are quite long and need to be summarized.

3-      Implications for policy/practice are different from the conclusion section and can be added at the end of the discussion section.

Author Response

  1. Summary 

We would like to thank the reviewer for taking the time to provide feedback on our manuscript. Please find our detailed response to specific comments below. 

  1. Questions for General Evaluation

Questions for general evaluation

Reviewer’s evaluation

Response and revisions

Does the introduction provide sufficient background and include all relevant references?

Can be improved

Based on the comment below, we have reduced text in the Introduction section by 28 lines and nearly 350 words.

Is the research design appropriate?

Yes

We would like to thank the reviewer for their kind feedback.

Are the methods adequately described?

Yes

Are the results clearly presented?

Yes

Are the conclusions supported by the results?

Can be improved

We have revised the Discussion and Conclusions sections as suggested per comment 3 below.

  1. Point-by-point response to Comments and Suggestions for Authors

Comment 1: I appreciate the authors for their time and efforts to revise the manuscript. However, some issues still need further attention.

1-      The main headings of the structured abstract (background, methods, results, conclusions) are missing.

Response 1: Per the IJERPH author instructions, the abstract should be “a single paragraph and should follow the style of structured abstracts, but without headings.” We believe the abstract follows these instructions, with the background covered in lines 13-18, methods in lines 18-23, results in lines 23-26, and conclusions in lines 26-28. 

Comment 2: 2-      The introduction section and its subsections are quite long and need to be summarized.

Response 2: We have revised the Introduction to substantially reduce the text, reducing the section by 28 lines and nearly 350 words (dropping from 161 lines and 1,906 words to 133 lines and 1,567 words). We hope these revisions strike a balance between streamlining the section and providing sufficient background on the programs being studied to be useful for practitioners reading the manuscript.

Comment 3: 3-      Implications for policy/practice are different from the conclusion section and can be added at the end of the discussion section.

Response 3: We have added a paragraph to the end of the Discussion section (page 13, lines 449-460) to discuss implications of the study, and removed implications from the Conclusions section.

  1. Response to Comments on the Quality of English Language

Response 1: We have made no major changes to the quality of the English language, based on the reviewer’s selection of “The quality of English does not limit my understanding of the research.”